# Overexpression of *LjPLT3* Enhances Salt Tolerance in *Lotus japonicus*

**DOI:** 10.3390/ijms24065149

**Published:** 2023-03-08

**Authors:** Jiao Liu, Leru Liu, Lu Tian, Shaoming Xu, Guojiang Wu, Huawu Jiang, Yaping Chen

**Affiliations:** 1Key Laboratory of South China Agricultural Plant Molecular Analysis and Genetic Improvement and Guangdong Provincial Key Laboratory of Applied Botany, South China Botanical Garden, Chinese Academy of Sciences, Guangzhou 510650, China; 2South China National Botanical Garden, Guangzhou 510650, China; 3University of Chinese Academy of Sciences, Beijing 100049, China

**Keywords:** *Lotus japonicus*, polyol/monosaccharide transporter LjPLT3, salinity stress, nitrogenase activity

## Abstract

Intracellular polyols are used as osmoprotectants by many plants under environmental stress. However, few studies have shown the role of polyol transporters in the tolerance of plants to abiotic stresses. Here, we describe the expression characteristics and potential functions of *Lotus japonicus* polyol transporter LjPLT3 under salt stress. Using *LjPLT3* promoter-reporter gene plants showed that *LjPLT3* was expressed in the vascular tissue of *L. japonicus* leaf, stem, root, and nodule. The expression was also induced by NaCl treatment. Overexpression of *LjPLT3* in *L. japonicus* modified the growth rate and saline tolerance of the transgenic plants. The *OELjPLT3* seedlings displayed reduced plant height under both nitrogen-sufficient and symbiotic nitrogen fixation conditions when 4 weeks old. The nodule number of *OELjPLT3* plants was reduced by 6.7–27.4% when 4 weeks old. After exposure to a NaCl treatment in Petri dishes for 10 days, *OELjPLT3* seedlings had a higher chlorophyll concentration, fresh weight, and survival rate than those in the wild type. For symbiotic nitrogen fixation conditions, the decrease in nitrogenase activity of *OELjPLT3* plants was slower than that of the wild type after salt treatment. Compared to the wild type, both the accumulation of small organic molecules and the activity of antioxidant enzymes were higher under salt stress. Considering the concentration of lower reactive oxygen species (ROS) in transgenic lines, we speculate that overexpression of *LjPLT3* in *L. japonicus* might improve the ROS scavenging system to alleviate the oxidative damage caused by salt stress, thereby increasing plant salinity tolerance. Our results will direct the breeding of forage legumes in saline land and also provide an opportunity for the improvement of poor and saline soils.

## 1. Introduction

Soil salinity is a major agricultural problem, being one of the most damaging abiotic stresses. High salinity leads to hyperosmotic conditions, which impede the ability of a plant to absorb water and nutrients from the soil, impacting seed germination, root, and shoot development and even yield [1]. Legumes have the ability to form a symbiotic relationship with nitrogen-fixing rhizobial bacteria to form nodules, which is helpful in the improvement of barren soil [2]. Legumes, such as *Medicago truncatula* and *Lotus japonicus*, are forage crops with high nutritional value and multiple ecological functions. However, salt stress is a major agricultural concern, as it negatively impacts plant growth, yield, and even the symbiotic relationship between legumes and rhizobia [3,4,5]. Therefore, increasing the salt tolerance of forage legumes is necessary not only for the improvement of soil but also to broaden the utilization rate of saline soil [6].

In general, to survive the effects of abiotic stresses, plants have developed complex and dynamic systems involving a wide range of biochemical and physiological processes [7]. The intracellular accumulation of compatible solutes functioning as osmoprotectants, such as polyols, is an important response mechanism of several plants to drought and salinity [8,9]. To date, thirty polyols identified in higher plants have been proven to function as important antioxidants and osmoprotectants in abiotic and biotic stress resistance. For example, mannitol accumulation is a crucial mechanism for salt/osmotic stress tolerance, heat stress-induced oxidative damage, and excessive solar irradiance [8,10,11]. Sorbitol synthesis/accumulation contributes to salt and drought tolerance in tomatoes, grapes, peaches, and persimmon trees [8]. In legumes, the synthesis and accumulation of polyols, such as mannitol, pinitol, and ononitol, increased under salt stress [12,13,14]. Ectopic expression of the mannose-6-phosphate reductase (M6PR) gene from celery in Arabidopsis and overexpression of the mtlD gene in transgenic Populus tomentosa improves salinity tolerance through the accumulation of mannitol [15,16]. Overexpression of myoinositol methyltransferase (IMT) and D-ononitol epimerase (GmOEP) from Glycine max in Arabidopsis increased salt and drought tolerance through the accumulation of D-ononitol and D-pinitol production, respectively [17,18]. Polyol transporters (PLTs) have been reported to participate in intercellular and interorgan communication of polyols in the plant; such as, PLTs in source leaves and phloem have been known to participate in long-distance transport in planta [19], while LjPLT11 (AM084328) was reported to facilitate intracellular translocation of pinitol inside the *L. japonicus* nodules [20]. Although PLTs have been identified and characterized in many plants [21,22,23,24,25,26,27,28,29], the roles of PLTs in plant salt tolerance remain unclear to date.

We previously reported that LjPLT11 regulates its substrate pinitol to maintain osmotic balance and stabilize the symbiosome membrane during nodule development in *L. japonicas* [20]. However, among 14 LjPLTs in *L. japonicus*, only the expression of LjPLT3 (*BT146435.1*) which belongs to the same subgroup as LjPLT11 was significantly induced by salt treatment [30]. Here, we analyzed pLjPLT3: GUS transgenic plants to characterize the expression pattern of LjPLT3 in *L. japonicus*. Our data show that overexpression of LjPLT3 in *L. japonicus* increased salinity tolerance through the reduction of oxidative damage.

## 2. Results

### 2.1. The Expression Pattern of LjPLT3 in L. japonicus

To investigate the expression pattern of LjPLT3 in *L. japonicus*, pLjPLT3: GUS plants were constructed and four transgenic lines with similar expression patterns were obtained. In 90% of pLjPLT3: GUS plants, GUS staining was detected in different tissues including in nodules, the stele of roots, leaves, flowers, stamens, and pods (Figure 1A–F). Transverse sections and microscope observations were used to determine further the expression of LjPLT3. The results showed that LjPLT3 was expressed in parenchyma cells and vascular bundles of leaves (Figure 1G), stems (Figure 1H), roots (Figure 1I), and nodules (Figure 1J). GUS signals were also observed in the infected zone of nodules which overlapped with Bacteroides expressing DsRed (Appendix A).

*pLjPLT3: GUS* plants were treated with 200 mM NaCl for 2 days before GUS staining was carried out. The increased expression of LjPLT3 in both leaves and roots in different lines was observed (Figure 1K). This result is consistent with that of LjPLT3 expression induced by salt treatment [30].

### 2.2. Overexpression of LjPLT3 Represses Plant Growth and Reduces Nodule Number in L. japonicus 

To investigate the functions of LjPLT3 in *L. japonicus*, transgenic plants overexpressing LjPLT3 (OEPLT3) were constructed. Fifteen OE lines were generated and three homozygous OEPLT3 lines (OE-9, OE-15, and OE-17) in which LjPLT3 expression was significantly stronger than in the wild-type plants were chosen for further research in this study (Appendix A). Plants were grown in symbiotic nitrogen fixation conditions and the phenotypes were observed at 4 weeks after inoculation with M. loti MAFF303099. 

The observation showed that the plant height and dry weight of transgenic plants were significantly reduced compared to the WT (Table 1 and Appendix A).

Then the nodulation phenotype was investigated. Compared to the wild type, there were significantly fewer nodules in OE plants, but the nodule size and the nitrogenase activity in the unit weight of nodules did not change (Table 1 and Appendix A).

To detect whether the developmental phenotype of transgenic plants was affected by symbiotic nitrogen-fixing nodules, three-day-old seedlings were grown in nitrogen-sufficient conditions for 4 weeks. Similarly, reduced plant height was also detected in OEPLT3 plants (Table 1 and Appendix A). These results indicated that the reduction in plant size of OEPLT3 plants was due to the high expression level of LjPLT3, rather than symbiotic nitrogen-fixing nodules. 

### 2.3. OEPLT3 Plants Displayed Increased Salinity Tolerance 

In view of the increased expression of LjPLT3 in response to the salt treatment [31], we tested the salinity tolerance of OEPLT3 plants at different developmental stages. Young seedlings were grown on ½ strength Broughton & Dilworth medium containing 5 mM KNO_3_ with different salt concentrations. Following the treatment of three-day-old seedlings with 100 mM NaCl for 10 days, the concentrations of chlorophyll were measured. The results showed that the total chlorophyll concentration in OEPLT3 leaves had not changed significantly, but that there was a dramatic decrease in wild-type leaves (Figure 2B). The survival rate of the wild type on 200 mM NaCl medium was only 31.5%, while that of OELjPLT3 plants was around 62.3–88.6% (Figure 2A,C). 

We also measured the salinity tolerance of mature seedlings. Two-week-old plants were treated with 150 mM NaCl (acclimation) and the growth parameters were evaluated 30 days later. The results showed that a sharp decrease of biomass was present in the wild type but that there was no significant difference in the OEPLT3 plants under salt stress (Figure 2D). 

Root nodules are more sensitive than the host plant itself to salinity stress [31]. Three-day-old plants inoculated with rhizobia for 4 weeks were treated with 100 mM NaCl and then the nitrogenase activity of nodules was tested 7 days after treatment (DAT). The results from the acetylene reduction assays showed that nitrogenase activity in wild-type nodules decreased by about 48%, while that in OELjPLT3 nodules decreased by about 29% (Figure 2E). Therefore, overexpression of LjPLT3 in *L. japonicus* increased the activity of nitrogenase under salt stress.

### 2.4. An Increase of Osmoregulation Substance in OEPLT3 Plants under Salt Stress

Compared to the wild type, OEPLT3 plants accumulated more soluble sugar in young leaves (the leaves in the second and third leaf position, from top to bottom) at both the end of the light period (EOL) and the end of the dark period (EOD) when plants were grown in nitrogen-sufficient conditions for 8 weeks (Figure 3A,B). The concentration of soluble sugar in OELjPLT3 roots was also higher than that in the wild-type roots (Figure 3C).

Given that proline and sugar products can be used as markers of environmental stress [4,32], concentrations of these small organic molecules were measured at 6 DAT. There were no significant differences in proline and sugar concentrations between different genotypes in the control group. The increased concentration of proline in OELjPLT3 leaves was greater than that in the wild type at 6 DAT as predicted (Figure 3D). The accumulation of glucose, fructose, sucrose, trehalose, ononitol, and pinitol in OELjPLT3 leaves reached higher levels than those in the wild type at 6 DAT as well (Figure 3E), which further confirmed that overexpression of LjPLT3 resulted in more accumulation of osmoregulation substances in *L. japonicus* under salt stress. 

### 2.5. OEPLT3 Plants Exhibit Increased Activity of Reactive Oxygen Species (ROS) Scavenging Mechanisms under Salt Stress

Increased salt tolerance was hypothesized to produce ROS that damaged seedlings, so H_2_O_2_ and O_2_^−^ concentrations were measured, given the increased salinity tolerance in OELjPLT3 lines. The results indicated that OEPLT3 displayed a 54.51–72.37% reduction in the level of H_2_O_2_ and a 33.33–36.32% reduction in the level of O_2_^−^ in leaves, compared to the wild type after NaCl treatment at different times (Figure 4A,B). Diaminobenzidine (DAB) and nitroblue tetrazolium (NBT) histochemical staining, which were used to determine the H_2_O_2_ and O_2_^−^ concentrations, gave consistent results with the determination of ROS concentration (Figure 4C,D).

To investigate why there was less ROS accumulation in OEPLT3 seedlings than in the wild type, we measured ROS scavenging-related enzyme activity and gene expression. The enzyme activity and gene expression were not significantly different between either plant type at 0 DAT. Although a trend of increasing enzymatic activity, such as ascorbate peroxidase (APX), superoxide dismutase (SOD), catalase (CAT), and peroxidase (POD), with NaCl treatment, was observed in the wild-type, activity remained noticeably lower than in the OEPLT3 leaves at 3 DAT (Figure 5A). Similarly, the transcription levels of APX, SOD, and CAT were higher in OELjPLT3 leaves than in the wild type at 3 DAT (Figure 5B).

Taken together, the strong ROS scavenging ability in the OEPLT3 seedlings resulted in less ROS accumulation in the transgenic plants than in the wild type.

## 3. Discussion

### 3.1. LjPLT3 Is a Salt Stress Response Gene in L. japonicus 

During abiotic stress, such as salt and osmotic stress, the rate of polyols synthesis increases in plants with the consequent induction of polyols transporter genes. For instance, mannitol accumulation is a crucial mechanism for salt/osmotic stress tolerance in Olea europaea [8]. Sorbitol synthesis/accumulation was also increased under salty conditions in peaches, persimmon trees, tomatoes, and Arabidopsis [28,33,34]. In grapes, the concentrations of mannitol, sorbitol, galactinol, myoinositol, and dulcitol were significantly increased in berry mesocarps in response to drought stress. The accumulation of pinitol, ononitol, and some sugars in response to saline stress was also promoted in *L. japonicus* (Figure 3E). With the increase in sugar and polyols concentrations, the expression of genes encoding polyol transporters is also induced. In rice, a Golgi localized monosaccharide transporter OsGMST1 and four polyol transporters (OsPLT3, OsPLT4, OsPLT13, and OsPLT14) significantly increased under salt osmotic and drought stress [35,36]. Salt and drought treatments significantly induced OeMaT1 expression in Olea europaea [8]. Similarly, the expression level of a mannitol transport gene (AtPLT6) was increased by salt stress in Arabidopsis [16]. Therefore, the transport and distribution of sugar alcohols/sugars in plants might also play important roles in response to salt stress. We previously identified 14 PLT genes in *L. japonicus* and reported that only LjPLT3 was induced significantly by NaCl treatment [30]. Here, GUS staining in pLjPLT3: GUS plants showed that GUS activity in both cotyledon and root increased when plants were exposed to saline treatment (Figure 1K), which indicates that LjPLT3 is a type of polyol transporter gene for salt stress in *L. japonicus*.

### 3.2. LjPLT3 Plays a Positive Role in L. japonicus Response to Salt Stress

In light of the importance of polyols in abiotic stress response, previous studies have shown that overexpression of related genes in plants by genetic engineering to improve the concentration of sugar alcohols/sugars can improve plant salt tolerance [12,13,14,15,16,17,18,37]. However, overexpression of sugar/sugar alcohol transporters may also reduce the salt tolerance of plants. Ectopic expression of a fructose and glucose/H^+^ symporter hexose transporter (MdHT2.2) of apple in tomato reduced ROS scavenging ability and thus reduced the salt tolerance of seedlings [1]. The result indicates that the intracellular distribution of sugar alcohols/sugars resulting from MdHT2.2 in tomatoes is not conducive to the elimination of intracellular ROS. In our work, Overexpression of LjPLT3 in *L. japonicus* increased sugar metabolism, thus inhibiting plant growth. Furthermore, OELjPLT3 lines displayed increased salt tolerance at both the seedling and mature stages (Figure 2), suggesting that LjPLT3 plays a positive role in *L. japonicus’* response to salt stress. Our data also imply that the transport of sugar alcohols/sugars in the plant caused by LjPLT3 could be beneficial for osmotic regulation and ROS scavenging, providing a novel genotype with great potential to be used as forage on salt-affected lands. 

### 3.3. Overexpression of LjPLT3 Increased Salt Tolerance by Reducing the ROS Damage in L. japonicus 

An increase in osmolytes correlates to a decrease in ROS production, thus improving tolerance to salt stress [38,39]. A large number of small molecules, such as proline and sugars/ sugar alcohols, will be produced to maintain cell osmotic potential in the plant in response to salt stress [1,4,40]. Here, OELjPLT3 plants accumulated more proline, trehalose, pinitol, and ononitol than the wild type after salt treatment (Figure 3). In addition to the maintenance of osmotic pressure, these osmoregulation substances can also be directly used as scavengers of ROS [41]. Therefore, we surmise that overexpression of LjPLT3 in *L. japonicus* might regulate osmoregulation substances to alleviate the damage resulting from oxidative stress, and thus improve the salt tolerance of plants.

Apart from osmotic adjustment compounds, antioxidant enzymes are also activated to eliminate excess ROS and thus prevent oxidative damage under abiotic stress [42]. Overexpression of MdHT2.2 in tomatoes produced lower antioxidant activities and higher ROS accumulation which caused salinity sensitivity [1]. In legume plants, such as *L. japonicus* and soybean (Glycine max), oxidative damage induced by abiotic stress is the result of an excess of ROS production [4,43]. Compared to the wild type, OELjPLT3 leaves accumulated less H_2_O_2_ and O_2_^−^ under salinity stress (Figure 4), while the activity of antioxidant enzymes and the expression of APX, SOD, and CAT in OELjPLT3 leaves increased to a higher level than those in the wild type after NaCl treatment (Figure 5). Therefore, the increased sugars/ sugar alcohols resulting from the overexpression of LjPLT3 in *L. japonicus* might eliminate excess ROS by activating the ROS scavenging mechanism, so as to maintain the balance of ROS and reduce oxidative damage caused by salt stress.

## 4. Materials and Methods

### 4.1. Plant Growth Conditions and Treatments 

*L. japonicus* “Miyakojiama” MG-20 was used as the wild type (WT) in this analysis. Seeds of MG-20 were collected and scarified with sulfuric acid for 5 min before being sterilized in 2% (*v*/*v*) sodium hypochlorite for 10 min and pre-germinated at 4 °C for 12 h. For inoculation with rhizobia, seedlings were grown in vermiculite pots containing 1/2 B&D [44] and infected with an A600 of 0.01 cell resuspension solution of Mesorhizobium loti (strain MAFF303099).

For symbiotic conditions, 3-day-old seedlings were inoculated with M. loti for 4 weeks, and then 10–15 independent plants from each line were measured for shoot height, dry weight, and nodule number. The biggest nodule from 15–20 independent plants were measured for nodule size. Pink nodules from 20–25 plants in different lines were collected for ARA. For nitrogen-sufficient conditions, 3-day-old seedlings were irrigated with Hoagland’s solution once a week and, 4 weeks later, 15–20 independent plants from each line were measured for shoot height and dry weight.

For NaCl treatment of seedlings, three-day-old, sterilized seedlings were transferred to square Petri dishes filled with 1/4 B&D containing 5 mM KNO_3_ and different concentrations of NaCl medium. For NaCl treatment under symbiotic conditions, plants were subjected to salinity stress by adding 150 mM NaCl to the 1/4 B&D solution at 28 days post-inoculation with M. loti. Nodules were collected for acetylene reduction assays 10 days after NaCl treatment. Control plants were maintained in a 1/4 B&D solution. 

For NaCl treatment under nitrogen-sufficient conditions, seedlings were grown in vermiculite pots containing Hoagland’s solution for 2 weeks [45]. Then control plants were irrigated with a nutrient solution without salt, while for saline treatment, plants were irrigated with a nutrient solution plus different concentrations of NaCl. For the measurement of physiologic indexes, plants were irrigated with 150 mM NaCl and leaves were harvested 6 days after treatment. For biomass assay, in order to avoid any osmotic shock due to salt treatment, plants initially received 50 mM NaCl and the concentration was then increased step-wise for 1 week (acclimation) until the final 150 mM concentration was reached [46].

### 4.2. Plasmid Constructs and Plant Transformation

Based on the promoter scan, several elements of TATABOX3 are located 270–500 nucleotides upstream of the transcription start site (https://www.dna.affrc.go.jp/PLACE/?action=newplace (accessed on 20 February 2020)). Therefore, for histospecificity expression analysis, a 2051 bp-fragment upstream of the initiation codon of LjPLT3 was amplified from genomic DNA. After digestion with Pst I/ EcoR I, the amplified fragments were cloned into pCAMBIA1391Z to construct the pLjPLT3: GUS plasmid. For the construction of the overexpression vector, the LjPLT3 coding sequence was amplified from the cDNA of *L. japonicus*. After digesting with restriction enzymes, the LjPLT3 sequence was cloned into the BamH I/ Hind III sites of pCAMBIA 1301 and driven by LjUbiquitin (LjUbi) promoter. These constructed vectors were transformed into MG-20 plants mediated by Agrobacterium tumefaciens strain AGL1 to gain stable transgenic plants following the method described in [47]. The homozygous transgenic lines were screened with hygromycin and the seeds from homozygous lines were collected and used for further studies. The primers used in the above constructs are listed in Appendix A.

### 4.3. GUS Histochemical Staining Analysis

For the analysis of tissue-specific expression, pLjPLT3: GUS transformed plants were infiltrated in X-Gluc staining solution (1 mM X-Gluc; 0.01 M phosphatic buffer solution, pH 7.0; 1 mM EDTA-Na_2_, pH 8.0; 0.5 mM K_3_ [Fe(CN)_6_]; 0.5 mM K_4_ [Fe(CN)_6_]) with vacuum treatment for 30 min, then incubated at 37 °C for 3–4 h and cleared with 70% (*v*/*v*) ethanol. Strained tissues of roots, nodules, stems, and leaves were selected and prepared in semi-thin sections (50 μm) with a vibrating-blade microtome (Leica (Wetzlar, Germany), VT1200S).

### 4.4. RNA Isolation and Expression Analysis

To identify the expression of LjPLT3 overexpression (OEPLT3) lines, total RNA from OEPLT3 lines was isolated by HiPure Plant RNA Mini Kit (Magen, R4151-02), and first-strand cDNAs were synthesized using GoScript TM Reverse Transcription System (Promega, Madison, WI, USA) as per the instructions. Primers of semi-quantitative PCR of LjPLT3 and LjATPase (control) using identifications are listed in Appendix A.

### 4.5. Analysis of Soluble Sugar and Starch

Soluble sugar concentration was analyzed using the colorimetric method with sulfuric acid-phenol as largely described in [48]. Samples (100 mg for fresh weight or 10 mg for dry weight) of WT and OEPLT3 lines were collected and extracted three times with 5 mL 80% (*v*/*v*) ethanol at 80 °C for 30 min. For each sample, all the extractives were incorporated and made up with 80% ethanol to 25 mL before 0.5 mL of the extractive was added to 1 mL of 0.9% (*v*/*v*) phenol and 5 mL of sulfuric acid in a reaction tube. Mixtures were reacted at room temperature for 30 min and A485 was measured with a spectrophotometer. The concentration of soluble sugar was calculated using a glucose standard curve. 

### 4.6. Measurement of Chlorophyll and Proline Concentrations

Measurements of total chlorophyll concentration were carried out using the method described by Arnon (1949) [49] with some modifications. Fresh leaf samples (100 mg) were collected and ground in 10 mL of 80% acetone. Absorbance was read at 645 and 663 nm using a spectrophotometer.

Free proline concentrations in the wild-type and transgenic plants were measured using the method described by Díaz et al. (2010) [50] with modifications. Fresh leaf samples of 0.5 g weight were extracted in 10 mL of 3% (*w*/*v*) sulphosalicylic acid (0.01 g/ 0.5 mL) and the homogenate was obtained through filtration using filter paper (Whatman No. 1). Toluene (8 mL) was added and left at room temperature for 30 min. The supernatant was collected, and absorbance was obtained at 520 nm using a spectrophotometer.

### 4.7. Antioxidant Analysis 

DAB and NBT staining was carried out to analyze the H_2_O_2_ and O_2_^−^ concentrations, respectively [20]. Fresh leaves were vacuum-infiltrated with 0.1 mg/mL DAB in 50 mM Tris–acetate buffer (pH 5.0) or 0.1 mg/mL NBT in 0.1 mM potassium phosphate buffer (pH 7.6) for 30 min, then incubated for 12 h at room temperature in the dark. Leaves were destained with 95% ethyl alcohol at 80 °C for 2 h and photographed. The H_2_O_2_ and O_2_^−^ concentrations were measured following the instructions in the kit (BC3595, BC1290. Solarbio, Beijing, China). 

The determination of enzyme activity was carried out using a method described by Wang et al. (2020). The activity of CAT was analyzed with the reaction mixture (50 mM potassium phosphate pH 7.0 and 10.5 mM H_2_O_2_) by measuring the initial linear rate of the decrease in absorbance at 240 nm. POD activity was analyzed with the reaction mixture (50 mM potassium phosphate pH 7.0, 9 mM guaiacol, and 19 mM H_2_O_2_) by measuring the formation of tetraguaiacol at 470 nm. SOD activity was measured by its ability to inhibit the photoreduction of nitroblue tetrazolium (NBT). APX activity was analyzed with the reaction mixture (50 mM potassium phosphate pH 7.0, 0.5 mM ascorbic acid, and 0.1 mM H_2_O_2_) by measuring the initial linear rate of the decrease in absorbance at 290 nm. One unit of APX activity was defined as 1 mg protein catalytic oxidizing 1 μM AsA in 1 min. One unit of SOD activity was defined as the amount of enzyme that inhibits 50% NBT photoreduction. One unit of CAT activity was defined as 1 mg protein catalytic decomposing 1 μM H_2_O_2_ in 1 min. One unit of POD activity was defined as 1 mg protein catalytic producing 1 μM tetraguaiacol in 1 min.

### 4.8. Acetylene Reduction Assay 

The method used for acetylene reduction assays was as previously described [46]. Fresh nodules of WT and OEPLT3 plants were collected into 10 mL headspace vials 8 weeks after inoculation (WAI). 300 μL of headspace was removed and replaced with 300 μL of acetylene in each vial. Mixed samples were incubated at 28 °C for 1 h and then 800 μL of headspace sample was used for quantification of produced ethylene using gas chromatography (Agilent, 7890A, Santa Clara, CA, USA).

### 4.9. Date and Statistical Analysis

The number of independent repetitions of experiments have been reported, together with the results of significance tests, in the figures. For all assays, differences between samples or time points were regarded as significant if *p* ≤ 0.05 and highly significant if *p* ≤ 0.01. All statistical analyses were carried out using SPSS Statistics V19.0. 

## Figures and Tables

**Figure 1 ijms-24-05149-f001:**
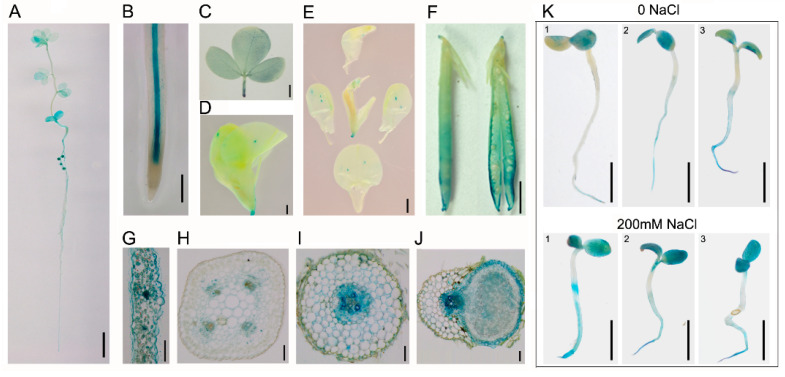
GUS staining of pLjPLT3: GUS plant. (**A**) GUS staining of a 4-week-old pLjPLT3: GUS seedling. (**B**) The root of a pLjPLT3: GUS plant showing GUS activity in the central cylinder. (**C**) Leaf showing GUS activity in the vascular bundle. (**D**,**E**) Flower of a pLjPLT3: GUS plant showing GUS activity in petiole and stamens. (**F**) Pod of a pLjPLT3: GUS plant. (**G**–**I**) Cross section of leaf, stem, and root showing the expression of LjPLT3 mainly in vascular bundles and parenchyma cells. (**J**) Cross section of nodule showing GUS activity in the cortical cell, vascular bundles, and infection zone. (**K**) GUS activity in pLjPLT3: GUS seedlings under NaCl treatment. 1, 2, 3 means different pLjPLT3: GUS lines. Scale bar = 1 cm (**A**,**K**), 0.5 cm (**B**,**C**), 0.2 cm (**D**–**F**), 250 μm (**G**,**I**,**J**), 100 μm (**H**).

**Figure 2 ijms-24-05149-f002:**
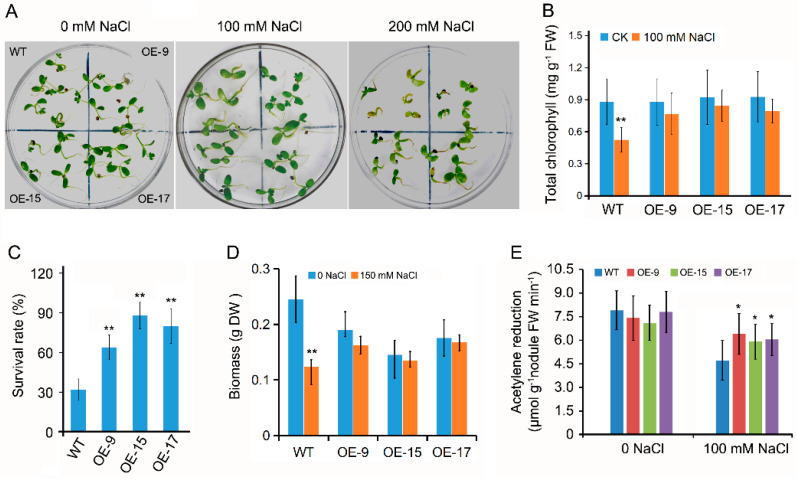
Salt stress tolerance evaluation of OELjPLT3 plants. (**A**) The phenotype of seedlings treated with different concentrations of NaCl. Three-day-old seedlings grew on 1/4 B&D medium with 100 or 200 mM NaCl, photographs taken 7 days later. (**B**) The chlorophyll concentration of seedlings shown in (**A**) under 100 mM NaCl treatment. (**C**) The survival rate of seedlings shown in (**A**) under 200 mM NaCl treatment. (**D**) The biomass of plants after 200 mM NaCl treatment (acclimation). Treatments were applied to ten-day-old plants for 30 days. (**E**) Results of the acetylene reduction assay after 100 mM NaCl treatment. Three-day-old seedlings were inoculated with M. loti. 100 mM NaCl treatments were applied after inoculation of rhizobia for 4 weeks and then nodules were collected 1 week later. 15–20 plants in different lines were used in (**B**,**C**). 4–6 plants in different lines were used in (**D**). Error bars represent the SD of three biological replicates. The significant differences were assessed using one-way ANOVA (*, *p* < 0.05; **, *p* < 0.01).

**Figure 3 ijms-24-05149-f003:**
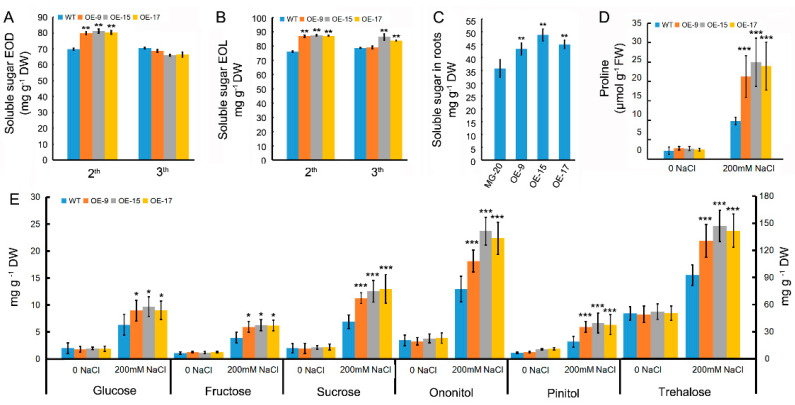
The changes of sugars/sugar alcohols in OELjPLT1 plants. (**A**,**B**) Soluble sugar concentration of leaves at the end of the day period (**A**, EOD) and end of the light period (**B**, EOL). (**C**) Soluble sugar concentration in roots of eight-week-old plants. (**D**) Proline concentration in wild type and OELjPLT3 leaves after NaCl treatment. (**E**) The concentration of sugars/sugar alcohols in the wild type and OELjPLT3 leaves after NaCl treatment. Leaves of eight-week-old plants were collected from the indicated leaf position from top to bottom in A and B. Seven-week-old plants were irrigated with 200 mM NaCl and leaves were collected for the measurement of proline, sugars, and polyols in D and E. Error bars represent the SD of three biological replicates. The significant differences were assessed using one-way ANOVA (*, *p* < 0.05; **, *p* < 0.01; ***, *p* < 0.001).

**Figure 4 ijms-24-05149-f004:**
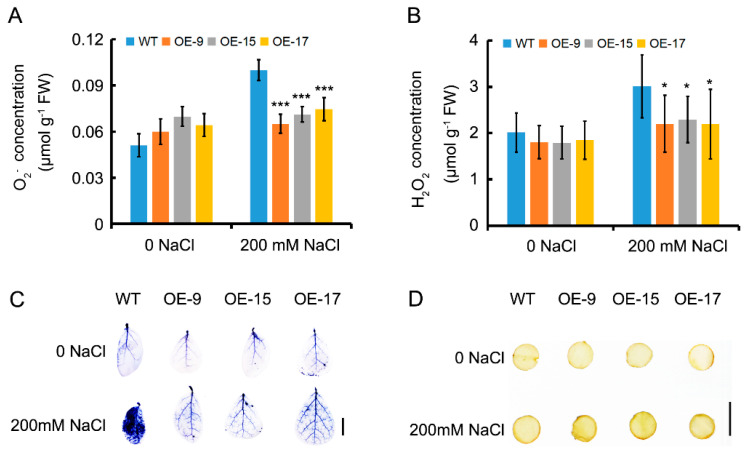
Changes in ROS concentration after salt treatment. (**A**) NBT staining of leaves. (**B**) DAB staining of leaves. (**C**) O_2_^−^ concentration in the leaves. (**D**) H_2_O_2_ concentration in the leaves. Three-week-old seedlings were treated with 200 mM NaCl and leaves were collected for staining and measurement of ROS 6 days after treatment. Error bars represent the SD of six biological replicates. Scale bar = 0.5 cm in (**A**,**B**). The significant differences were assessed using one-way ANOVA (*, *p* < 0.05; ***, *p* < 0.001).

**Figure 5 ijms-24-05149-f005:**
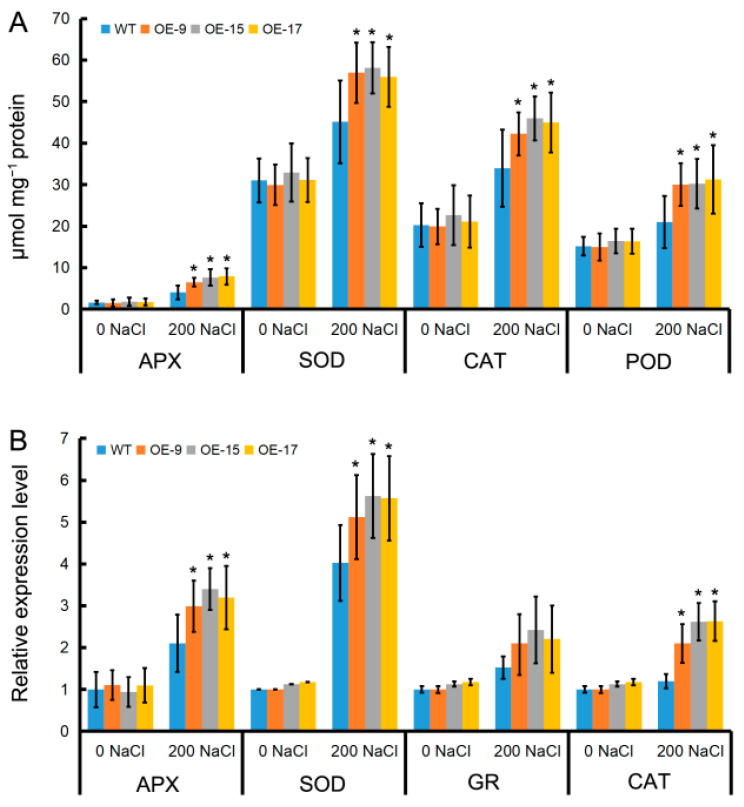
The effect of LjPLT3 on the ROS scavenging mechanism after salt treatment. (**A**) The activities of the antioxidant enzymes after 200 mM NaCl treatment. (**B**) Expression levels of genes encoding ROS scavenging enzymes after NaCl treatment. The expression of LjAPX, LjSOD, LjGR, and LjCAT was calculated relative to that of LjACTIN and the expression in the wild type without NaCl treatment was set to 1. Three-week-old seedlings were treated with 200 mM NaCl and leaves were collected for the analysis of the enzyme activity and gene expression 6 days after 200 mM NaCl treatment. Error bars represent the SD of 5–6 biological replicates. The significant differences were assessed using one-way ANOVA (*, *p* < 0.05).

**Table 1 ijms-24-05149-t001:** Phenotypic characteristics of OEPLT3 plants.

Phenotype	WT	OE-9	OE-15	OE-17
Symbiotic Condition				
Height (cm)	8.92 ± 0.57	8.34 ± 0.62 *	7.82 ± 0.41 **	7.91 ± 0.32 **
Dry Weight (g)	0.15 ± 0.03	0.12 ± 0.06 *	0.09 ± 0.01 **	0.10 ± 0.01 **
Nodule number (per plant)	19.31 ± 4.12	17.01 ± 6.51 *	14.02 ± 3.28 **	14.95 ± 3.02 **
Nodule Size (cm)	0.15 ± 0.01	0.15 ± 0.02	0.16 ± 0.05	0.15 ± 0.01
Acetylene reduction assay (µmol C_2_H_4_ h^−1^ g FW^−1^)	1.51 ± 0.38	1.51 ± 0.25	1.48 ± 0.49	1.49 ± 0.57
Nitrogen-Sufficient Condition				
Height (cm)	12.15 ± 2.31	11.54 ± 2.14 *	9.67 ± 1.59 **	9.86 ± 1.64 **
Dry weight (g)	0.24 ± 0.06	0.21 ± 0.06 *	0.15 ± 0.03 **	0.18 ± 0.04 **

Data represent the Mean ± SD of three independent biological replicates. Asterisks indicate significant differences with Student’s *t*-test (* indicates *p* ≤ 0.05, ** indicates *p* ≤ 0.01).

## Data Availability

The data presented in this study are available in this article and Appendix A.

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
