# Peer review of "Overexpression of LjPLT3 Enhances Salt Tolerance in Lotus japonicus"

_ijms, 2023, doi:10.3390/ijms24065149_

Round 1

Reviewer 1 Report

The authors shoud clarificate/correct some inaccuracies between main text and figures and their legends (marked in attached file) and add methods used for DsRed experiment. In lines 294 and 385 authors claim increased GR expression in transgenic plants, however, judging from fig.5 it is not significantly changed but not mentioned SOD activity is elevated. Other comments are in atatched file.

Author Response

The authors shoud clarificate/correct some inaccuracies between main text and figures and their legends (marked in attached file) and add methods used for DsRed experiment. In lines 294 and 385 authors claim increased GR expression in transgenic plants, however, judging from fig.5 it is not significantly changed but not mentioned SOD activity is elevated. Other comments are in attached file.

Response:

Dear reviewer, thank you for your nice suggestions!

The inaccuracies between main text and figures and their legends has been corrected based on your comments. And all the suggestions you mentioned in the attached file were also revised.

The expression of GR did not have significant change and we deleted the description of GR here in the revised manuscript.

We have updated Figure 2 and Figure 4 with the correct Line number or Figure caption.

And all the revised parts were marked in the red color in the new version.

Reviewer 2 Report

This manuscript is about functional verification of lotus PLT3 gene. There are two major experiments: (1) the promoter induction of PLT3 (2) overexpressing of PLT3. Impressively, they also did a lot of work to examine phenotypes under abiotic stresses and physiological changes like sugar, ROX. The manuscript is well organized and the English wiring is fluent and concise. In addition, it is a good topic to study the function of PLT3 gene.

Here are some suggestions and comments:

(1)    Introduction, L66-69

It is too brief to introduce the transporter protein PLTs. Please expand these sentences.

(2)    L414

“a 2051 bp-fragment upstream of the initiation codon of LjPLT3”. Do you want to say “translation start site”. So the 2051-bp must include 5’ UTR. How length of the 5’ UTR it is? If you don’t know the transcription start site. A prediction of promoter structure is necessary, by showing core cis-elements in this region. Why not scan the promoter regions and display important cis-elements regarding salt induction.

(3)    Fig 1k

The results are very clear. What is the meaning “1,2,3”? Just repeats? Please put Fig.1k into a box to give the boundary of k.

(4)    Table 1

WT=wild type. Why not use empty-vector plants as control? It is normal to find regeneration plants showing abnormal growth. What is the generation of these transformed plants? T2? T3?

(5)    Section 2.3

How many plants did you used to get the data of Figure 2?

(6)    Fig 3B

The column. It is truth there is significant difference between WT and OE-9 at 3th?

Fig2

(7)    Fig5

It is clear transgenic plants have higher enzyme activity of these antioxidant enzymes. The authors want to say higher enzyme activity, better tolerance. From some researches, we can see” higher enzyme activity, lower tolerance”. How to explain? Any discussion?

Author Response

This manuscript is about functional verification of lotus PLT3 gene. There are two major experiments: (1) the promoter induction of PLT3 (2) overexpressing of PLT3. Impressively, they also did a lot of work to examine phenotypes under abiotic stresses and physiological changes like sugar, ROX. The manuscript is well organized and the English wiring is fluent and concise. In addition, it is a good topic to study the function of PLT3 gene.

Here are some suggestions and comments:

(1)    Introduction, L66-69

It is too brief to introduce the transporter protein PLTs. Please expand these sentences.

 Response: Thank you for your nice comments. We expanded a little bit about PLT as a transporter in plant as “Polyol transporters (PLTs) have been reported to participate in intercellular and interorgan communication of polyols in the plant. Such as, PLTs in source leaves and phloem have been known to participate in long-distance transport in planta [19], while LjPLT11 was reported to facilitate intracellular translocation of pinitol inside the L. japonicus nodules [20].” In the revised manuscript.

(2)    L414

“a 2051 bp-fragment upstream of the initiation codon of LjPLT3”. Do you want to say “translation start site”. So the 2051-bp must include 5’ UTR. How length of the 5’ UTR it is? If you don’t know the transcription start site. A prediction of promoter structure is necessary, by showing core cis-elements in this region. Why not scan the promoter regions and display important cis-elements regarding salt induction.

Response: Thank you for your nice comments. Based on the promoter scan, several elements of TATABOX3 is located 270-500 nucleotides upstream of the transcription start site (https://www.dna.affrc.go.jp/PLACE/?action=newplace). In the revised manuscript, we added the sentence.

(3)    Fig 1k

The results are very clear. What is the meaning “1,2,3”? Just repeats? Please put Fig.1k into a box to give the boundary of k.

Response: Thank you for your nice comments. We put Fig.1k into a box in the revised manuscript. 1, 2, 3 means different pLjPLT3: GUS lines, we have already added this sentence in the Figure legend.

(4)    Table 1

WT=wild type. Why not use empty-vector plants as control? It is normal to find regeneration plants showing abnormal growth. What is the generation of these transformed plants? T2? T3?

Response: Thank you for your nice comments. empty-vector plants could produce other phenotypes, so we used MG-20 as the WT here. We used the third generation of transformed plants in this study and all the three OEPLT3 lines were homozygous.

(5)    Section 2.3

How many plants did you used to get the data of Figure 2?

Response: Thank you for your nice comments. We have used 15-20 seedlings for chlorophyll determination and also survive rate count. Here we added a sentence in the legend of Fig.2 as “15–20 plants in different lines were used in (B) and (C). 4-6 plants in different lines were used in (D).”

(6)    Fig 3B

The column. It is truth there is significant difference between WT and OE-9 at 3th?

Response: Thank you for your careful review! Yes, there is no significant difference between WT and OE-9. Now we replaced Fig 3 with a revised version.

(7)    Fig5

It is clear transgenic plants have higher enzyme activity of these antioxidant enzymes. The authors want to say higher enzyme activity, better tolerance. From some researches, we can see” higher enzyme activity, lower tolerance”. How to explain? Any discussion?

Response: Thank you for your nice comments. The antioxidant enzyme system of different plants could increase or decrease at different degrees in response to different stresses, but here, overexpression of LjPLT3 might reduce the oxidative damage caused by salt stress via the increased enzyme activity.

Reviewer 3 Report

Overall written with adequate reporting of results, however statistic outcome statements are needed for all ANOVA results to be inserted within the describing paragraph that goes with each Figure.  References support results, and provided good background understanding of the objectives.

Page 1. Recommend add NCBI Accession number LjPLT3 mRNA in Keywords.

 Page 2- Lines 70 and 79, need to ADD-NCBI Accession numbers for LjPLT3 and LjPLT11 mRNA’s used for expression analyses.

RESULTS section:  All paragraphs in RESULTS section, that are reporting results and linked to a Figure (2,3,4,5 that shows ANOVA data. Should include the specifics of the ANOVA analyses results. 

  The captions on Figures are OK, as written, once this additional information is added into corresponding text paragraph.

Page 4-5. Text section 2.3.

Page 5. In the TEXT paragraph (not Figure caption that is OK) but in descriptive paragraph that reports results of FIG.2. there should be a statistic outcome statement similar to: One-way ANOVA showed there was statistically significant differences in acetylene levels  between group means of treatments, NaCL with 100 mM NaCl as determined by one-way ANOVA, F(3,20) = 252.44, P < 0.01; α = 0.05) (this is an example authors will add in the real ANOVA values).    

Also, The error bars in the 100 NaCl treatment are confusing as they do not appear to be different statistically from the WT plants, and show strong overlap, which would suggest they are NOT different from the WT control.

Page 5. Line 209. The last sentence is a discussion statement suggest reword to a results statement.   “Overexpression of....L. japonicus increased the activity of...... “

Page 5. Line 216. Need to reword sentence to results statement,  ‘Overexpression of .....in L. japonicus increased sugar metabolism, thus......”

 Page 6. Line 214  Section 2.4. Should add in TEXT paragraph (not Figure caption that is OK) but in descriptive paragraph that reports results of FIG.3. there should be a statistic outcome statement similar to: One-way ANOVA..showed there was statistically significant differences in acetylene levels  between group means of treatments, NaCL with 100 mM NaCl as determined by one-way ANOVA, F(3,20) = 252.44, P < 0.01; α = 0.05).    

Page 6. Line 259. "gave consistent results" ? Explain in percentages or other numerical summary statement.

Author Response

Dear Review,

Thank you so much for your nice comments. We have already revised the manuscript based on your suggestions and all the revised parts were marked in red color in the new version.

Overall written with adequate reporting of results, however statistic outcome statements are needed for all ANOVA results to be inserted within the describing paragraph that goes with each Figure.  References support results, and provided good background understanding of the objectives.

Page 1. Recommend add NCBI Accession number LjPLT3 mRNA in Keywords.

Response: Thank you for the comment. NCBI Accession number (BT146435.1) was added following LjPLT3 in the keywords.

 Page 2- Lines 70 and 79, need to ADD-NCBI Accession numbers for LjPLT3 and LjPLT11 mRNA’s used for expression analyses.

Response: Thank you for the comment. NCBI Accession number BT146435.1 for LjPLT3 and AM084328 for LjPLT11 were added in Line 70-79. The added number was marked in red color.  

RESULTS section:  All paragraphs in RESULTS section, that are reporting results and linked to a Figure (2,3,4,5 that shows ANOVA data. Should include the specifics of the ANOVA analyses results. 

The captions on Figures are OK, as written, once this additional information is added into corresponding text paragraph.

Response: Thank you for your nice comments. The description of Figures in the corresponding text was revised and marked in the red color in the edited manuscript.

Page 4-5. Text section 2.3.

Page 5. In the TEXT paragraph (not Figure caption that is OK) but in descriptive paragraph that reports results of FIG.2. there should be a statistic outcome statement similar to: One-way ANOVA showed there was statistically significant differences in acetylene levels  between group means of treatments, NaCL with 100 mM NaCl as determined by one-way ANOVA, F (3, 20) = 252.44, P < 0.01; α = 0.05) (this is an example authors will add in the real ANOVA values).    

Also, the error bars in the 100 NaCl treatment are confusing as they do not appear to be different statistically from the WT plants, and show strong overlap, which would suggest they are NOT different from the WT control.

Response: Thank you for your nice comments. We revised the caption of Figure 2.

Page 5. Line 209. The last sentence is a discussion statement suggest reword to a results statement.   “Overexpression of....L. japonicus increased the activity of...... “

Page 5. Line 216. Need to reword sentence to results statement,  ‘Overexpression of .....in L. japonicus increased sugar metabolism, thus......”

Response: Dear reviewer, many thanks for your suggestions. The text section about line 209 and line 216 have been modified.

Page 6. Line 214  Section 2.4. Should add in TEXT paragraph (not Figure caption that is OK) but in descriptive paragraph that reports results of FIG.3. there should be a statistic outcome statement similar to: One-way ANOVA..showed there was statistically significant differences in acetylene levels  between group means of treatments, NaCL with 100 mM NaCl as determined by one-way ANOVA, F(3, 20) = 252.44, P < 0.01; α = 0.05).   Page 6. Line 259. "gave consistent results" ? Explain in percentages or other numerical summary statement.

Response: Thank you for your nice comments. We added the reduction of ROS level in percentages in the text as “The results indicated that OEPLT3 displayed 54.51%-72.37% reduction in the level of H2O2 and 33.33%-36.32% reduction in the level of O2- in leaves, compared to the wild type after NaCl treatment at different times (Fig. 4A, B).”. The revised part was marked in red.